# Adversaries Fight Back: Revisiting Test-Time Defenders of Vision-Language Model under Adaptive Attacks

## Abstract

Recent works have introduced test-time defenses to protect zero-shot classification of vision-language models (VLM) against adversarial attacks. While these methods claim to achieve test-time efficiency and performance comparable to train-time defenses on CLIP, we evaluate 6 state-of-the-art VLM test-time adversarial defenses and find that their robustness significantly overestimated when subjected to defense-specific attacks (adaptive attacks). By simply applying basic adaptive attacks including full-defense gradient and approximated-gradient attacks, we find that the robustness for half of the evaluated methods can be largely reduced. We then further develop a flexible Distributional-Difference-Targeting Adaptive Attack (CLIP-DDT) to enhance the strength of adversarial points for CLIP's test-time defenses, even when full-defense gradient adaptive attack is hard to compute or ineffective. Specifically, we identify a ubiquitous but vulnerability-inducing key step in these defenses: the zero-shot measurement of a detectable distributional difference between adversarial and clean data. Our method targets this Achilles' Heel by operating on this distributional difference during attacking the underlying static model when basic adaptive attacks fail or it can be combined with basic adaptive attacks for boosted effect otherwise. Experiments across 13 datasets demonstrate that our method generates strong adversarial samples that further degrade the robustness of various test-time defenses, revealing a false sense of security in CLIP's zero-shot robustness.

## 1 Introduction

The advent of large-scale pre-trained vision-language model (VLM) has generated significant interest in the research community (Jia et al., 2021; Ramesh et al., 2021; Li et al., 2022). A prominent example, CLIP (Radford et al., 2021), has the ability to perform zero-shot classification by matching image features with prompted text embeddings of class names. Despite its widespread adoption as a foundation model in many security-sensitive applications, the adversarial robustness of CLIP has been shown to be weak (Shayegani et al., 2023; Zhao et al., 2023). Consequently, improving the robustness of CLIP's zero-shot classification (zero-shot robustness) has become a critical problem studied in many previous works (Li et al., 2024; Zhang et al., 2024a; 2025; Mao et al., 2023).

Initial efforts (Li et al., 2024; Zhang et al., 2024a; Mao et al., 2023; Schlarmann et al., 2024; Wang et al., 2024) to solve this problem drew inspiration from the adversarial training (AT) paradigm, and tackled this problem from the perspective of train-time adversarial fine-tuning or prompt tuning. Though AT has been proven successful in traditional single-modal classification, applying adversarial finetuning to CLIP presents two primary drawbacks: overfitting to the fine-tuning dataset and loss of real-world knowledge.

To circumvent these issues, researchers have started to explore a new paradigm: test-time defenses (Wang et al., 2025; Sheng et al., 2025; Xing et al., 2025; Tong et al., 2025; Zhang et al., 2025). These test-time defenses achieve reasonable robustness without significantly sacrificing clean accuracy or VLM's generalization abilities. Moreover, they maintain an inference time comparable to the original CLIP model. This efficiency presents a notable advantage over traditional test-time defenses

(Nie et al., 2022; Zhang et al., 2024b), which often require hundreds or even thousands of times more computing time during inference.

The growing prominence of test-time defenses for zero-shot robustness raises a critical question: Is it possible that the adversarial robustness in these methods is overestimated? In traditional classification, it is well-established by previous works (Crose et al., 2022; Carlini et al., 2019; Athalye et al., 2018; Tramer et al., 2020) that many test-time defenses are vulnerable against defense-specific attacks (adaptive attacks); a defense that cannot withstand such scrutiny is considered of limited utility. To our knowledge, however, no such systematic research has been conducted for test-time defenses in the context of VLM zero-shot classification. To answer the question above, we take one step further: Is there a generalizable way to construct stronger adversarial samples for different test-time VLM defenses? A generalizable attack framework would allow the identification of common vulnerabilities and provide a more rigorous perspective for assessing the robustness of newly proposed VLM test-time defenses.

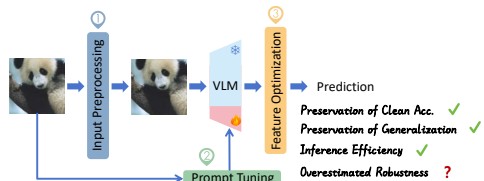

Figure 1: The 3 categories of test-time defense for CLIP. The first category purifies the inputs before they are fed to the VLM. The second category adapts the model prompt to fit the test data. The third category optimizes the feature in the direction of removing adversarial traits.

To explore these questions, we first take a closer look at 6 newly proposed state-of-the-art test-time VLM adversarial defenses (Wang et al., 2025; Sheng et al., 2025; Xing et al., 2025; Tong et al., 2025; Zhang et al., 2025) spanning the defense paradigms of input purification, feature space purification, and test-time prompt tuning (shown in Figure 1). A review of the original papers reveals a significant evaluation gap: Among these works, TTC (Xing et al., 2025), R-TPT (Sheng et al., 2025) and AOM (Tong et al., 2025) provide no evaluation against adaptive attacks, while CLIPure-Cos (Zhang et al., 2025), CLIPure-Diff (Zhang et al., 2025) and TAPT (Wang et al., 2025) only report results against ineffective ones. In this paper, we surprisingly find that the robust accuracy for 3 of these 6 methods can be dramatically reduced by basic adaptive attacks (i.e. full-gradient and approximated-gradient). To further assess the worst-case robustness of the remaining defenses, we propose a flexible method to enhance the adaptive attack strength, applicable to CLIP's test-time defenses that are differentiable, approximately differentiable, or contain non-differentiable components. Our approach is based on a key insight: these defenses all contain a key step that explicitly or implicitly measures or utilizes a detectable distributional difference between adversarial and clean data, inducing a potential vulnerability itself. By identifying this step, our method adjust this detectable difference while conducting basic adaptive attack, depending on its availability.

The proposed key-step targeting adaptive attack offers a twofold benefit. First, it provides a generalizable strategy against a wide array of defenses. While test-time defenders can arbitrarily combine non-differentiable or computationally complex components to evade basic full-gradient adaptive attacks, our method provides a way to construct simple and effective attacks. Attacking such a common foundational step also aligns with the principle of pursuing simplicity in attack design (Tramer et al., 2020). Second, even when a defense if differentiable, it may contain components that are unnecessary for attacks, while essential for the defense. A direct full-gradient attack can thus become trapped in local optimal, limiting its effectiveness. Our method also helps mitigate this issue.

It is important to note that instead of challenging the validity of current VLM test-time defenses, we view the insights from these methods as a significant step towards achieving true zero-shot robustness. This paper seeks to contribute to this goal by providing some suggestions for future test-time defenders and adaptive attackers. The main contributions of this paper are:

- We evaluate 6 newly proposed test-time defense methods for CLIP's zero-shot robustness from Wang et al. (2025); Sheng et al. (2025); Xing et al. (2025); Tong et al. (2025); Zhang et al. (2025) and find that their robustness is largely overestimated by applying basic adaptive attacks including full-defense or approximated gradient attacks (Crose et al., 2022).

- We propose a method named as CLIP-DDT to boost the attacking strength against VLM test-time defenses via exploiting a key step that exists ubiquitously within them. This approach is still effective even when the full or approximated gradient attack fails or is

hard to compute. We also offer suggestions for future works in this area for both defenders and attackers.

- Experimental results demonstrate that our method effectively reduces the robustness of different test-time VLM defenses across 13 datasets.

## 2 RELATED WORKS

**Train-Time Adversarial Defenses of VLM** Given the success of adversarial training (AT) in traditional classification, some researchers (Li et al., 2024; Zhang et al., 2024a; Mao et al., 2023; Schlarmann et al., 2024; Wang et al., 2024) have transformed this idea into adversarial fine-tuning for pretrained VLM models. TeCoA (Mao et al., 2023) proposes to use prompted texts for adversarial data generation. PMG-AFT (Wang et al., 2024) adds two additional learning objectives as regularization to keep the generalization capability of the original CLIP. FARE (Schlarmann et al., 2024) proposes an unsupervised way for adversarial sample generation to alleviate the problem of overfitting and clean accuracy degradation. To relieve the problem of heavy computation in AT, APT(Li et al., 2024) and AdvPT (Zhang et al., 2024a) propose to tune the prompts only.

**Test-Time Adversarial Defenses of VLM** Researchers have found that train-time adversarial tuning may devastate the real-world knowledge and generalization abilities in large-scale pre-trained models, thus turning to test-time defenses. Among test-time defenses, TTC (Xing et al., 2025) follows the paradigm of input preprocessing and purifies the input image by applying counter-attack perturbation. R-TPT (Sheng et al., 2025) and TAPT (Wang et al., 2025) follow the paradigm of test-time prompt tuning with different learning objectives. CLIPure-Cos (Zhang et al., 2025), CLIPure-Diff (Zhang et al., 2025) and AOM (Tong et al., 2025) follow the paradigm of feature purification by using likelihood estimation and Gaussian noise. Test-time defenses have multiple advantages compared to the training-time methods as in Figure 1 and we will focus on their robustness. A more detailed review for these methods will be provided later.

**Adaptive Attacks** Adaptive attacks are the adversarial attacks crafted to attack specific defense (Crose et al., 2022; Carlini et al., 2019; Athalye et al., 2018; Tramer et al., 2020). Previous research has shown that some test-time defenses can be broken by specially tuned adaptive attacks, highlighting the importance of understanding the limitations of newly proposed defenses. Although no standard evaluation procedure is proposed, (Tramer et al., 2020) suggests striving for simplicity in the attack design to relieve the optimization problem and (Crose et al., 2022) recommends using full-gradient and approximated-gradient attack for defense assessment. In this work, we provide a generalizable method that further boosts the adaptive-attack strength in the CLIP-based zero-shot classification.

## 3 PRELIMINARIES

### 3.1 BASIC FORMULATION

Given the image feature encoder $f$ and text encoder $g$, CLIP calculates the logits of image $x$ belonging to different classes as:

$$\text{logit}_{c_i}(x) = \cos\left(f_\theta(x), g_\phi(c_i)\right), \tag{1}$$

where $f_\theta$ is the vision encoder, $g_\phi$ is the text encoder, and $c_i$ is the class name text of the candidate class prompted by a template that can take the form of "photo of [class]".

In adversarial attacks a bounded perturbation $\delta$ is added to a clean testing sample $x$ with ground truth class $T(x)$ to generate an adversarial sample $x_{\text{adv}}$ by:

$$x_{\text{adv}} = \text{clamp}(x + \delta_{\text{adv}}, 0, 1) \tag{2}$$

where

$$\delta_{\text{adv}} = \arg\max_\delta \mathcal{L}(\text{logit}(x + \delta), T(x)) \text{ s.t. } ||\delta||_p \leq \epsilon. \tag{3}$$

To achieve zero-shot robustness, test-time defenses generally span three techniques targeting different steps of classification, which are input purification, test-time prompt tuning, and feature optimization. When we are given a new test-time defense for VLM, we can generally write the predicted

result for class $c_i$ as:

$$\text{logit}_{c_i}(x_{\text{adv}}) = \cos(\tilde{f}_\theta(x_{\text{adv\_purified}}), g_{\phi\_\text{tuned}}(c_i)) \tag{4}$$

where $x_{\text{adv\_purified}}$ is the preprocessed input, $\tilde{f}_\theta(x)$ is the optimized image feature and $g_{\phi\_\text{tuned}}$ is the text encoder with the tuned prompts.

## 3.2 BRIEF REVIEW OF STATE-OF-THE-ART TEST-TIME DEFENSES

To demonstrate the overestimated robustness in test-time defenses for CLIP, in this section we first give a review for 6 state-of-the-art defenses from 5 papers (Wang et al., 2025; Sheng et al., 2025; Xing et al., 2025; Tong et al., 2025; Zhang et al., 2025):

**TTC** TTC (Xing et al., 2025) is built on the observation that the features of adversarial samples are trapped in a falsely stable area that makes the feature less sensitive to tiny random noise. Thus if the ratio of feature drift caused by tiny noise is larger than a threshold, the image feature undergoes an purification process that add a counter-attack perturbation that leads to largest distortion in a self-supervised way and then used in prediction.

**CLIPure-Cos and CLIPure-Diff** CLIPure-Cos and CLIPure-Diff are two versions of CLIPure (Zhang et al., 2025). The rationale beneath this defense is that adversarial data is moved away from the clean data manifold and usually have a likelihood lower than clean ones. Thus a feasible method to purify the feature is to optimize its likelihhod. While CLIPure-Diff relies on an extra DifussionPrior network to estimate feature likelihood, CLIPure-Cos uses cosine similarity between the image feature and the blank template to estimate the likelihood of the feature to achieve unprecedented purification efficiency. Except from effectively increasing robustness, another benefit of this likelihood optimization is that there is no negative effect on clean accuracy.

**AOM** Inspired by previous works in random smoothing (Cohen et al., 2019; Levine & Feizi, 2020; Schoten et al., 2024), AOM (Tong et al., 2025) proposes to use Gaussian noise to relieve the adversarial effects of inputs. Specifically, Gaussian noise is added to inputs multiple times to calculate an averaged feature, which is supposed to be less adversarial and closer to the clean data distribution. Then the averaged feature is used as an anchor point and AOM applies one step gradient descent towards the direction of away from the original adversarial feature to move the test feature closer to the clean distribution. The final prediction is calculated with this updated anchor feature.

**R-TPT** R-TPT (Sheng et al., 2025) is a defense method that follows the paradigm of test-time prompt tuning. The method is built on the observation that adversarial samples can lose their adversarial trait after data augmentations. Thus it applies AugMix (Hendrycks et al., 2020) to the input 64 times and the augmented views with largest confidence are selected to tune the model prompt with the pointwise entropy minimization objective. Since the adversarial views generally has smaller entropy, their effects in entropy-minimizing prompt tuning are overcome by augmented samples with reduced adversarial trait. After the prompt tuning, the predictions of augmented views are selected and weighted based on the similarities between prediction pairs to filter out outlier predictions.

**TAPT** As another test-time prompt tuning based defense, TAPT (Wang et al., 2025) first generates augmented views of input image with random resized crop and horizontal flip, then tunes the prompt by augmented views with lowest entropy. However, unlike R-TPT, the averaged batchwise-entropy instead of the pointwise entropy is calculated as the tuning objective and the data augmentation here is differentiable. The method also uses an alignment of mean and variance statistics with clean and robustly tuned models. Since the code for this defense is not provided, in this paper we implemented TAPT-L, which is the language prompt tuning version of TAPT.

## 4 DISTRIBUTIONAL-DIFFERENCE TARGETING ADAPTIVE ATTACK

### 4.1 GENERAL IDEA OF CLIP-DDT

Given the variation in defense paradigms and techniques, previous work (Crose et al., 2022) finds it hard to provide standard instructions for constructing stronger adaptive attacks except the basic ones in traditional classification. However, it is suggested that basic adaptive attacks may not provide an effective evaluation for test-time defenses' worst-case robustness.

In this section, we propose CLIP-DDT to enhance the strength of basic full-defense gradient or approximated gradient adaptive attacks when they fail to attack CLIP's test-time defenses. As a start, one may notice that despite the apparent difference in their specific techniques, CLIP-based defense methods mentioned above all try to diminish the adversarial traits in a zero-shot way, which relies on adopting general adversarial assumptions to new test data. Therefore, there also exists a key step that the defense relies on to measure the assumed distributional difference between clean and adversarial samples of zero-shot datasets.

We hypothesis this step induces potential vulnerability itself, as we may explicitly craft adversarial samples to hide this discrepancy while still keeping the most of their attacking strength. Thus we consider adjusting this measured distributional difference during PGD-based attacks. Given a defense method, we inject this extra objective into the PGD optimization of basic adaptive attack based on its availability. By targeting this distributional difference measuring step, we also get rid of potential attack-impeding operations in the defense. The attacking objective of CLIP-DDT can generally be expressed as:

$$\mathcal{L}_{\text{CLIP-DDT}} = \mathcal{L}_{\text{CE}}(\text{logit}(x_{\text{CLIP-DDT}}), T(x)), \tag{5}$$

where

$$x_{\text{CLIP-DDT}} = \text{op}_{x_{\text{adv}}}(\mathcal{L}_{\text{dist-diff}}(x_{\text{adv}}, x_{\text{clean}})) \tag{6}$$

is the adaptive adversarial sample optimized by our method, $\mathcal{L}_{\text{dist-diff}}$ is the objective of measured distributional difference utilized by the defense. $\text{op}_{x_{\text{adv}}}()$ is the operation on $x_{\text{adv}}$ to adjust this targeted distributional difference in each iteration.

Please note that this specific operation is different for the targeting defenses (as demonstrated below), thus the specific formulation can have some variations. However, this idea of targeting the distributional difference key step is shared across the developed adaptive attacks. The general idea of our method is illustrated in Figure 2.

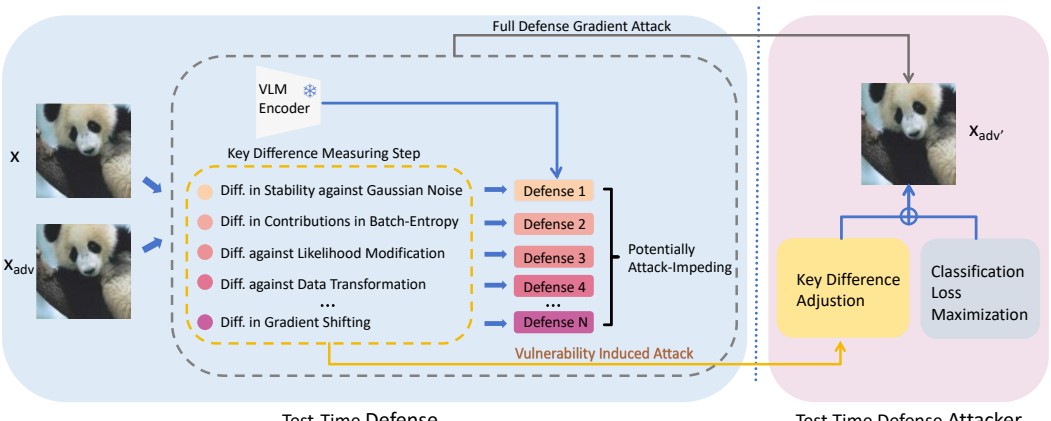

Figure 2: An illustration of the idea of our CLIP-DDT. Our method enhances the flexibility and strength of adaptive attacks by identifying the vulnerability-inducing key distributional difference step in a given test-time VLM defense and adjust this key difference during attack generation.

## 4.2 ANALYSIS AND ATTACK OF SOTA DEFENSES

In this section, we will use case studies to illustrate the vulnerability for some of these test-time defenses even against basic adaptive attacks and how to apply our CLIP-DDT to further reduce the robustness. In all the experimental results, Static stands for static model attack and Full stands for full-defense gradient attack.

**TTC : Difference in Gradient-Shifting.** To understand the feature drift distributional difference between adversarial and clean data utilized by TTC, one can use Taylor expansion to approximately express the feature distortion caused by tiny random noise as the dot product of the feature gradient on each dimensionality of input image and the added noise. As the attacking optimization pushes

Table 1: Zero-shot robust accuracies under PGD-10 attack with $\epsilon$=4.0/255 for defense method TTC.

| Attack | CIFAR10 | CIFAR100 | STL10 | Food101 | Oxford | Flowers102 | EuroSAT | TinyImageNe | ImageNet | Caltech101 | Caltech256 | StanfordCars | PCAM | Avg |
|--------|---------|----------|-------|---------|--------|------------|---------|-------------|----------|------------|------------|--------------|------|-----|
| Static | 31.2 | 12.3 | 52.0 | 16.4 | 21.3 | 14.6 | 5.6 | 5.2 | 19.6 | 56.2 | 34.0 | 9.2 | 52.4 | 25.4 |
| Full | 15.0 | 3.3 | 29.4 | 6.0 | 8.7 | 4.6 | 3.4 | 0.8 | 11.3 | 32.8 | 14.8 | 3.7 | 31.4 | 12.7 |
| **Ours** | **3.6** | **0.5** | **12.7** | **0.2** | **1.2** | **0.7** | **0.0** | **0.1** | **1.8** | **13.5** | **4.6** | **0.1** | **20.1** | **4.5** |

the features to adversarial distribution, the gradient of input image becomes saturated and it's harder to further change the feature in this rather smooth area. As a result, the dot product with tiny random noise component is small.

A natural way of attacking this defense is to take the full gradient of the counterattack noise applied to adversarial samples and let the adversaries to attack against the maximal distortion imposed. The original paper of TTC has proposed an approximated version of this attack because the original defense implementation contains non-differentiability. However, no result against this adaptive attack is given in the original paper. We first implement this approximate-gradient adaptive attack and find that it already reduces the reported robustness, shown as Full in Table 1.

To further enhance the strength of the adaptive attack, we identify the key distributional difference between adversarial and clean data measured in this defense: the different extent of gradient shifting from clean position. That is, the discrepancy between gradient shifted to saturated area for adversaries vs. the gradient stayed in the non-saturated area for clean ones is the key for the success of the defense. Therefore we consider the objective of explicitly minimizing this discrepancy when attacking the model. The most straightforward realization is to skip the later attacking steps once the attack is successful to minimize accumulated shifts of gradient. During the attack, we compute the maximum distortion injected adversarial data $x_{\text{CLIP\_DDT\_drift}}$ approximately as:

$$x_{\text{CLIP\_DDT\_drift}} \approx x_{\text{CLIP\_DDT}} + \delta^0 + \mu\nabla_\delta||f_\theta(x_{\text{CLIP\_DDT}} + \delta^0) - f_\theta(x_{\text{CLIP\_DDT}})||, \quad (7)$$

where $\delta^0$ is the randomly initialized noise. Then our attacking objective is:

$$\mathcal{L}_{\text{CLIP\_DDT}} = \mathcal{L}_{\text{CE}}(\text{logit}(x_{\text{CLIP\_DDT\_drift}}), T(x)) \cdot \mathbf{1}_{\text{argmax}(\text{logit}(x_{\text{CLIP\_DDT\_drift}})) \neq T(x)}. \quad (8)$$

Since the approximate-gradient attack is effective, we simply apply our operation to it to reduce the robustness when approximate-gradient adaptive attack fails. The further reduced robustness presented in Table 1 denoted as Ours shows the effectiveness of our method. In average, there is a robustness decline of 20.9% compared to static-model attack across the datasets.

Table 2: Zero-shot robust accuracies under PGD-10 attack with $\epsilon$=1.0/255 for defense methods CLIPure-Cos and CLIPure-diff.

| Defense | Attack | CIFAR10 | CIFAR100 | STL10 | Food101 | Oxford | Flowers102 | EuroSAT | TinyImageNe | ImageNet | Caltech101 | Caltech256 | StanfordCars | PCAM | Avg |
|---------|--------|---------|----------|-------|---------|--------|------------|---------|-------------|----------|------------|------------|--------------|------|-----|
| | Full | 92.1 | 72.4 | 98.6 | 89.5 | 84.1 | 76.1 | 42.5 | 60.1 | 86.4 | 99.5 | 94.8 | 54.7 | 63.6 | 78.0 |
| CLIPure-Cos | Static | 3.0 | 1.0 | 26.0 | 0.4 | 0.5 | 3.4 | 0.0 | 0.7 | 2.5 | 29.0 | 19.0 | 0.0 | 0.0 | 6.6 |
| | **Ours** | **2.3** | **0.4** | **20.5** | **0.4** | **0.0** | **1.8** | **0.0** | **0.1** | **1.6** | **20.6** | **14.7** | **0.0** | **0.0** | **4.8** |
| | Full | 94.4 | 76.1 | 96.1 | 95.3 | 77.1 | 83.5 | 88.6 | 66.7 | 86.9 | 82.5 | 95.6 | 54.9 | 65.5 | 81.8 |
| CLIPure-Diff | Static | 8.2 | 2.3 | 30.9 | 0.9 | **0.6** | 5.5 | 0.4 | 1.8 | 3.9 | 17.7 | 22.0 | 0.1 | 0.5 | 7.3 |
| | **Ours** | **7.1** | **0.9** | **27.5** | **0.4** | 0.8 | **3.5** | **0.0** | **1.3** | **3.1** | **12.8** | **20.9** | **0.0** | **0.0** | **6.0** |

Table 3: Zero-shot robust accuracies under PGD-10 attack with $\epsilon$=4.0/255 for defense methods CLIPure-Cos and CLIPure-diff.

| Defense | Attack | CIFAR10 | CIFAR100 | STL10 | Food101 | Oxford | Flowers102 | EuroSAT | TinyImageNe | ImageNet | Caltech101 | Caltech256 | StanfordCars | PCAM | Avg |
|---------|--------|---------|----------|-------|---------|--------|------------|---------|-------------|----------|------------|------------|--------------|------|-----|
| | Full | 86.6 | 64.2 | 98.4 | 88.8 | 80.1 | 74.7 | 35.7 | 54.9 | 87.2 | 99.0 | 94.8 | 56.1 | 55.4 | 75.1 |
| CLIPure-Cos | Static | 0.3 | 0.0 | 2.6 | 0.0 | 0.0 | 0.0 | 0.0 | 0.0 | 0.0 | 0.0 | 4.7 | 0.0 | 0.2 | 0.6 |
| | **Ours** | **0.3** | **0.0** | 2.7 | **0.0** | **0.0** | **0.0** | **0.0** | **0.0** | **0.0** | **0.1** | **4.3** | **0.0** | **0.2** | **0.6** |
| | Full | 92.7 | 72.0 | 96.1 | 93.6 | 76.2 | 81.7 | 88.0 | 65.2 | 86.9 | 85.9 | 95.4 | 54.7 | 58.8 | 80.1 |
| CLIPure-Diff | Static | 1.1 | 0.1 | 3.9 | 0.0 | 0.0 | 1.0 | 0.0 | 0.0 | 0.0 | 0.4 | 4.4 | 0.0 | 0.1 | 0.9 |
| | **Ours** | **1.1** | **0.0** | **3.3** | **0.0** | **0.0** | **0.0** | **0.0** | **0.0** | **0.0** | **0.0** | **4.2** | **0.0** | **0.0** | **0.7** |

**CLIPure-Cos and CLIPure-Diff: Difference in Stability against Likelihood Modification.** By optimizing the estimated likelihood of the test data, CLIPure-Cos and CLIPure-Diff achieve impressive robustness that is close to clean accuracy on 13 downstream datasets, showing a sign of unsuccessful adversarial attacks. To boost the adaptive attack strength, we analyze the key distributional difference utilized here: the difference in prediction stability after likelihood modification. That is, while clean features have stable predictions before and after this likelihood optimization, adversarial features are purified to the clean area, which produce correct predictions. A possible reason for the unsuccessful full-defense gradient attack is that the large feature oscillation caused by the whole feature update steps interferes with the adversarial update, as we find an interesting phenomenon that the static model attack, which is assumed to be weaker than adaptive attack dramatically reduces the robust accuracy.

To adjust this key distributional difference during attack without bringing too much difficulty to the attack update, we regularize each update iteration to preserve the adversarial trait before and after a single step of approximated likelihood modification. The attacking objective is:

$$\mathcal{L}_{\text{CLIP\_DDT}} = \mathcal{L}_{\text{CE}}(\text{logit}_o(x_{\text{p\_drift}}, x), T(x)), \tag{9}$$

where $\text{logit}_o$ is the predicted logits of original features before purification and

$$x_{\text{p\_drift}} \approx x_{\text{CLIP\_DDT}} + \delta^0 + \mu \nabla_\delta \log p(x_{\text{CLIP\_DDT}} + \delta^0), \tag{10}$$

where $\delta^0$ is the randomly initialized noise and $\log p$ is the likelihood estimated as in the paper of CLIPure:

$$\log p = cos(f_i, \bar{f}_t), \tag{11}$$

where $\bar{f}_t$ is the blank text template embedding and $f_i$ is the image embedding. We conduct experiments for these two defenses with $\epsilon = 4.0/255$ in Table 3 and $\epsilon = 1.0/255$ in 2. The results show the effectiveness of CLIP-DDT with suitable key distributional difference adjusting.

**AOM: Difference in Direction Changes against Gaussian Noise.** The whole defense is based on the assumption that the relative direction from adversarial feature to the Gaussion added one is a reasonable choice for gradient descent. The key distributional difference between adversarial and clean feature here is the relative direction change against the added Gaussian noise. Therefore we target this key distributional difference by limiting the effectiveness of this relative direction:

$$\mathcal{L}_{\text{CLIP\_DDT}} = \mathcal{L}_{\text{CE}}(\text{logit}(f_{\text{CLIP\_DDT\_Gaussian}} + \alpha \cdot (f_{\text{CLIP\_DDT\_Gaussian}} - f_{\text{CLIP\_DDT}})), T(x)), \tag{12}$$

where $f_{\text{CLIP\_DDT\_Gaussian}}$ is the Gaussian added image feature and $\alpha = 0.2$ is the hyperparameter in the original defense. In this case, our CLIP-DDT downgrades back to the form of basic full-gradient attack, which brings the robustness to around 0 with $\epsilon = 4.0/255$ in Table 4.

Table 4: Zero-shot robust accuracies under PGD-10 attack with $\epsilon$=4.0/255 for defense method AOM.

| Attack | CIFAR10 | CIFAR100 | STL10 | Food101 | Oxford | Flowers102 | EuroSAT | TinyImageNet | ImageNet | Caltech101 | Caltech256 | StanfordCars | PCAM | Avg |
|---|---|---|---|---|---|---|---|---|---|---|---|---|---|---|
| Static | 40.9 | 22.3 | 69.0 | 49.5 | 27.6 | 34.0 | 3.2 | 27.1 | 20.9 | 21.0 | 24.8 | 22.5 | 18.9 | 29.4 |
| **Ours (Full)** | **0.1** | **0.0** | **2.0** | **0.0** | **0.0** | **0.1** | **0.0** | **0.0** | **0.0** | **0.0** | **2.9** | **0.0** | **0.6** | **0.4** |

**R-TPT: Difference in Prediction variation with Augmentation.** For this multi-view prompt tuning based method, both the AugMix data augmentations and weighted prediction based on index-selection bring difficulties to full-defense gradient-based attack generation by making the defense non-differentiable. To construct adaptive attacks, we again identify the key distributional difference utilized here: the difference in prediction variation with augmentation imposed. That is, unstable predictions for adversarial samples vs. stable predictions for clean samples against augmentation. We restrict this difference by forcing the augmented-batch adversarial loss with differentiable random resized crop and flip operations in place of AugMix:

$$\mathcal{L}_{\text{CLIP\_DDT}} = \sum_{\text{aug}_i} \mathcal{L}_{\text{CE}}(\text{logit}(x_{\text{CLIP\_DDT\_aug}_i}), T(x)). \tag{13}$$

We conduct this CLIP-DDT attack when basic PGD attack fails and the robustness is shown as Ours in Table 5, which is close to 0 for $\epsilon$=4.0/255. Note that in this case we largely increase the strength of adaptive attacks without visiting the computational heavy prompt-tuning components, making our method more efficient.

Table 5: Zero-shot robust accuracies under PGD-10 attack with $\epsilon$=4.0/255 for defense method R-TPT. Note that PGD-Full-Defense is not applicable here due to the non-differentiability.

| Attack | CIFAR10 | CIFAR100 | STL10 | Food101 | Oxford | Flowers102 | EuroSAT | TinyImageNet | ImageNet | Caltech101 | Caltech256 | StanfordCars | PCAM | Avg |
|---|---|---|---|---|---|---|---|---|---|---|---|---|---|---|
| Static | 31.8 | 17.7 | 75.5 | 45.4 | 41.8 | 39.9 | 2.0 | 15.0 | 32.8 | 82.1 | 56.2 | 16.1 | 31.3 | 37.5 |
| **Ours** | **0.0** | **0.0** | **2.0** | **0.0** | **0.0** | **2.8** | **0.0** | **0.0** | **0.5** | **3.3** | **0.4** | **0.3** | **0.3** | **0.7** |

**TAPT: Difference in Contributions in Batchwise Entropy.** TAPT (Wang et al., 2025) is another test-time prompt tuning based defense. To generate adaptive attacks, we first notice that the whole process is differentiable and we can actually directly apply full-defense gradient that includes augmentation, entropy calculation and tuning process in backward propagation. We find that this full-gradient adaptive attack already largely reduces the robustness, as shown in Table 6 as Full.

To generate stronger attacks for TAPT, we observe that this defense contains two distributional differences between adversarial and clean data that we may utilize, namely the stability difference against data augmentation as in R-TPT (Sheng et al., 2025) and the contributional difference in the averaged batch-entropy. While the full-defense gradient attack already contains the operation of generating multiple augmented views and handles the stability difference to some extent, in this case we focus on the later difference. In TAPT, as the logits of clean augmented views outweigh that of remained adversarial views, the prompt is finetuned in the direction that benefits the confident classification of clean views. In order to break this defense, we add a simple objective that encourages small entropy of successful adversarial sample in augmented views during attack update. During the attacking process, the prompts are first tuned in the same way as in the defense:

$$
\theta_{\text{tuned}} = \theta + \nabla_\theta (\text{Batch\_Entropy}(\tilde{x}) +
$$
$$
\sum_{\theta' \in \theta_{\text{clean}}, \theta_{\text{robust}}} \mathcal{L}_1(\text{statistics}(f_\theta(\tilde{x})) - \text{statistics}(f_{\theta'}(\tilde{x})))/2, \tag{14}
$$

where $\tilde{x}$ is the augmented views of the input $x$, Batch-Entropy is the entropy calculated on averaged prediction of the batch. The objective to optimize the adversarial sample can then be expressed as:

$$
\mathcal{L}_{\text{CLIP\_DDT}} = \mathcal{L}_{\text{CE}}(\text{mean}_{\tilde{x}_{\text{CLIP\_DDT}}}(\text{logits}(\tilde{x}_{\text{CLIP\_DDT}}))) - \lambda \cdot \text{entropy}(\tilde{x}_{\text{CLIP\_DDT\_adv}}), \tag{15}
$$

where

$$
\tilde{x}_{\text{CLIP\_DDT\_adv}} = \cup \, \mathbf{1}_{\text{argmax}(\text{logits}_\theta(\tilde{x}_{\text{CLIP\_DDT}})) \neq T(x)} \cdot \tilde{x}_{\text{CLIP\_DDT}} \tag{16}
$$

are the augmented views of our optimized $x_{\text{CLIP\_DDT}}$ that successfully attack the model and $\lambda$ is the hyperparameter that balance the cross-entropy loss for attacking and entropy loss. This extra distributional-difference minimizing objective increases the role of adversarial views in averaged logit calculation to balance the beneficial effects of multiple views. We apply this adaptive attack when standard full-gradient attack fails to further reduce the robustness. The results are shown as Ours in Table 6. In average, there is a robustness decline of 38.0% across the datasets compared to basic static model attack. Note that unlike the case of RTPT, we use $\epsilon = 1.0/255$ in Table 6 as in the main setting of original paper. Results with $\epsilon = 4.0/255$ are in Appendix.

Table 6: Zero-shot robust accuracies under PGD-10 attack with $\epsilon$=1.0/255 for TAPT-L.

| Attack | CIFAR10 | CIFAR100 | STL10 | Food101 | Oxford | Flowers102 | EuroSAT | TinyImageNet | ImageNet | Caltech101 | Caltech256 | StanfordCars | PCAM | Avg |
|---|---|---|---|---|---|---|---|---|---|---|---|---|---|---|
| Static | 69.5 | 42.5 | 90.3 | 74.7 | 58.2 | 59.8 | 12.3 | 31.4 | 61.5 | 94.3 | 76.6 | 34.9 | 44.0 | 57.7 |
| Full | 12.5 | 8.2 | 52.8 | 21.7 | 21.3 | 32.4 | 0.5 | 4.9 | 27.4 | 64.5 | 34.4 | 9.0 | 17.4 | 23.6 |
| **Ours** | **9.9** | **7.1** | **50.5** | **16.1** | **15.0** | **28.3** | **0.0** | **3.8** | **23.3** | **57.6** | **30.8** | **6.9** | **7.0** | **19.7** |

The distributional difference adjusting operation on TAPT includes a hyperparameter $\lambda$ to trade-off between classification and entropy loss. In Figure 3 we conduct an ablation study $\lambda$ on robust accuracy of defense TAPT (Wang et al., 2025) with 500 data points of PCAM and OxfordPet. Note that $\lambda$ equals 0 means conducting one extra baseline attack without considering CLIP-DDT objective when the initial baseline attack fails. When $\lambda$ gets larger, The entropy regularization helps to distort the prompt tuning direction to enhance attack strength. As $\lambda$ gets too large, the entropy regularization starts to affect the adversarial objective.

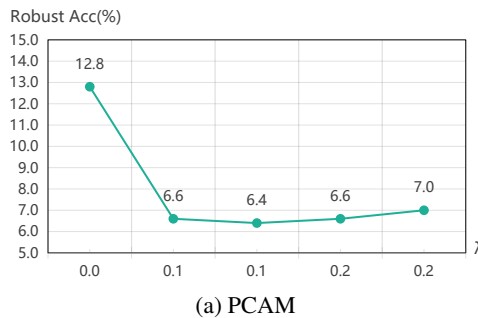
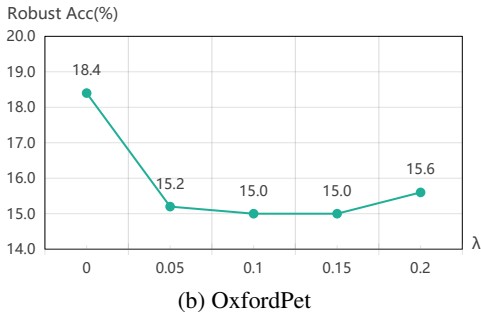

(a) PCAM                                                   (b) OxfordPet

Figure 3: The effect of CLIP-DDT hyperparameter $\lambda$ on robust accuracy of TAPT on 500 data points of (a) PCAM and (b) OxfordPet.

## 4.3 DISCUSSION

Test-time defense papers are usually evaluated in their own ways, making it hard to see their worst-case performance. We think one way to at least relieve the problem is to develop adaptive attacks with simple idea that can generalize to more test-time defenses and combined with basic adaptive attacks. Here we provide some suggestions both for CLIP's robustness defenders and adaptive attackers: 1) Test-time defenders may consider reducing the dependency on distributional differences that are not strongly-related to adversarial traits as these traits can be unstable against optimization in adversarial subspaces. 2) Test-time defenders may consider more aggressive preprocessing of data that is hard to approximate. The generalization abilities of large-scale pre-trained VLM makes this possible without severely harming performance. 3) Instead of always trying to attack the whole defense, adaptive attackers may consider attacking against the distributional difference measuring step in CLIP's test-time defenses as attacking this step either causes the prompts to be tuned in the wrong direction or make the model fail to identify the adversaries.

## 5 EXPERIMENTAL SETTINGS

We apply 10-step PGD (Madry et al., 2018) to attack the defenses with $\epsilon = 4.0/255$ and $\epsilon = 1.0/255$. The backbone model we use is ViT-B/32 for all papers except CLIPure (Zhang et al., 2025), which uses ViT-L/14 in the main experiments. To test the robustness, we randomly choose 1000 data points from each of 13 datasets following previous works FARE (Schlarmann et al., 2024) and CLIPure (Zhang et al., 2025). Note that some methods report the results on different splits of the dataset, thus there can be some fluctuations on the absolute values of the results. However, this difference doesn't affect the generalizability of the relative values of different attack methods. Additional data information is in Appendix.

Please note that for CLIPure-Cos, CLIPure-Diff, TTC and R-TPT, the original code is provided and we conduct our evaluation based on the original provided code. For AOM and TAPT, there is no publicly available code and we conduct the evaluation based on our implementation that generally has the same level of avgerage robustness. Our CLIP-DDT takes roughly 0.64s to generate adversarial sample for one test data, nearly the same as 0.62s for basic static attack on one A6000 GPU.

## 6 CONCLUSION

In this paper, we discuss the false sense of security in test-time defenses for CLIP's zero-shot classification. Despite the promising future of this defense paradigm, we find the adversarial robustness for many SOTA defenses vulnerable against basic adaptive attacks. We then develop a flexible adaptive attack method that can be combined with basic adaptive attacks and has the potential to be applied to evaluate the worst-case robustness of newly proposed test-time defenses. We hope our work can provide some insights for the development of zero-shot robustness.

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

# A APPENDIX

**Datasets** The datasets we used are the common ones in previous works and introduced in Appendix. which include CIFAR10 (Krizhevsky & Hinton, 2009), CIFAR100 (Krizhevsky & Hinton, 2009), STL10 (Coates et al., 2011),,Caltech101 (Li et al., 2006), Caltech256 (Li et al., 2006), OxfordPets (Parkhi et al., 2012), EuroSAT (Helber et al., 2019), PCAM (Bejnordi et al., 2017), Food101 (Bossard et al., 2014), Flowers102 (Nilsback & Zisserman, 2008), StanfordCars (Krause et al., 2013),TinyImageNet (Deng et al., 2009) and ImageNet (Deng et al., 2009).

**Additional Experimental Results** Here we present some additional experimental results on defense TAPT-L with attacking range $\epsilon = 4.0/255$:

Table 7: Zero-shot robust accuracies under PGD-10 attack with $\epsilon$=4.0/255 for defense method TAPT-L.

| Attack | CIFAR10 | CIFAR100 | STL10 | Food101 | Oxford | Flowers102 | EuroSAT | TinyImageNet | ImageNet | Caltech101 | Caltech256 | StanfordCars | PCAM | Avg |
|---|---|---|---|---|---|---|---|---|---|---|---|---|---|---|
| Static | 19.2 | 13.1 | 68.2 | 38.9 | 26.9 | 34.0 | 2.2 | 8.1 | 40.0 | 32.5 | 49.6 | 12.6 | 23.0 | 25.0 |
| Full | 0.0 | 0.0 | 0.4 | 0.0 | 0.0 | 2.2 | 0.0 | 0.0 | 0.3 | 1.9 | 1.9 | 0.1 | 0.0 | 0.5 |
| **Ours** | **0.0** | **0.0** | **0.4** | **0.0** | **0.0** | **1.7** | **0.0** | **0.0** | **0.1** | **0.6** | **1.4** | **0.0** | **0.0** | **0.3** |

