# OpenReview forum: "Adversaries Fight Back: Revisiting Test-Time Adversarial Defenders of Vision-Language Model under Adaptive Attacks"
_ICLR.cc/2026/Conference — Submitted to ICLR 2026_

### Official Review · Reviewer_y6QN · 2025-10-27

**Soundness:** 3
**Presentation:** 3
**Contribution:** 2
**Rating:** 4
**Confidence:** 4

**Summary:**

This study investigates the true robustness of recently proposed test-time defense methods for VLMs. While prior works claim strong zero-shot robustness, this paper shows that such claims are often overestimated when evaluated against naïve or non-adaptive attacks. The authors first re-evaluate six test-time defenses and demonstrate that their robustness drops dramatically under basic adaptive attacks using full or approximated defense gradients. To further strengthen adversarial evaluations, the paper introduces CLIP-DDT (Distributional-Difference Targeting Adaptive Attack), a generalizable adaptive attack framework that exploits a common vulnerability shared across these defenses, i.e., their reliance on distributional differences between clean and adversarial data. By directly manipulating this key distributional step during attack optimization, CLIP-DDT generates stronger adversarial examples that severely degrade the defenses’ robustness across 13 datasets. The results expose a false sense of security in current VLM test-time defenses and provide concrete guidance for developing more reliable robustness evaluation protocols.

**Strengths:**

- The study reveals a major evaluation flaw in the current literature on CLIP-based test-time defenses by showing that their reported robustness is significantly overestimated when not tested against adaptive attacks

- The authors then introduce CLIP-DDT (Distributional-Difference Targeting Attack), a conceptually simple yet powerful and generalizable adaptive attack framework that identifies and exploits a common vulnerability (distributional difference measurement) shared across diverse defense paradigms, making it applicable to both differentiable and non-differentiable defenses.

- Their comprehensive empirical validation demonstrates consistent robustness degradation and offers clear, reproducible evidence that advances understanding of zero-shot adversarial robustness in multimodal systems.

**Weaknesses:**

- The proposed Distributional-Difference Targeting Attack is primarily presented as an empirical framework without a formal theoretical analysis of why targeting distributional differences consistently weakens defenses. Without this, it is not certain if the proposed approach only works against the surveyed test time defenses, and might not actually scale to others. I would sugges a mathematical formulation explaining its convergence behavior or general conditions for its success (e.g., connection to gradient alignment or information geometry), at least to make the contribution more rigorous.

- Although the paper focuses on CLIP and its derivatives, it does not evaluate whether the same vulnerabilities persist in more recent or structurally distinct VLMs (e.g., BLIP, Flamingo, or LLaVA). The claim of generalizability across VLMs remains unproven outside the CLIP family, which undermines the broader applicability of the findings. Also the surveyed defenses are majorly focused on purification and prompt tuning approaches, but there are other approaches focused on detection [1, 2, 3, 4].

- The paper mentions future suggestions for test-time defenders but does not test or demonstrate any mitigations (e.g., randomized defenses, adaptive defenses, etc).

- The authors claim that CLIP-DDT is efficient and flexible but provides only a single runtime comparison (0.64s vs. 0.62s for static attacks). There is no analysis of scaling behavior across input size, model depth, or dataset volume.

**References**

[1] Fares, Samar et al. “MirrorCheck: Efficient Adversarial Defense for Vision-Language Models.” ArXiv abs/2406.09250 (2024): n. pag.
[2] Zhou, Qi et al. “Defending LVLMs Against Vision Attacks through Partial-Perception Supervision.” ArXiv abs/2412.12722 (2024): n. pag.
[3] Sun, Jiachen et al. “Safeguarding Vision-Language Models Against Patched Visual Prompt Injectors.” ArXiv abs/2405.10529 (2024): n. pag.
[4] Zhang, Xiaoyu et al. “JailGuard: A Universal Detection Framework for LLM Prompt-based Attacks.” (2023).

**Questions:**

See weaknesses

---

### Official Review · Reviewer_K95W · 2025-10-27

**Soundness:** 2
**Presentation:** 2
**Contribution:** 3
**Rating:** 4
**Confidence:** 3

**Summary:**

This paper presents a critical evaluation of recent test-time adversarial defenses for VLMs like CLIP. The authors argue that the adversarial robustness of several test-time defenses is significantly overestimated because they are not evaluated against adaptive attacks. To demonstrate this, the paper introduces a novel and flexible adaptive attack framework called CLIP-DDT.

**Strengths:**

1. Test-time defenses are gaining traction for their efficiency and ability to preserve model generalization. A thorough security audit of this emerging paradigm is both necessary and timely, providing a much-needed reality check for the community.

2. The paper is well-framed, tackling the critical issue of overestimated robustness in test-time defenses due to the omission of adaptive attacks.

3. The experimental setup is described in a detailed and thorough manner.

**Weaknesses:**

1. In the presence of both train-time defenses and test-time defenses, is the proposed adaptive attack method still effective, and does it still outperform the baseline?

2. Computational Overhead Analysis. The paper notes that CLIP-DDT takes roughly the same time as a basic static attack (0.64s vs 0.62s). However, the actual cost of an adaptive attack depends on the complexity of the defense's forward pass. For defenses that include internal optimization loops (e.g., AOM, TAPT), a more detailed breakdown of the total computational cost for end-to-end adversarial example generation would give a more accurate view of practical overhead.

3. How was the trade-off hyperparameter $\lambda $ chosen for the main experiments? Was a single value used across defenses/datasets, or was it tuned per setting? Please report sensitivity of CLIP-DDT to $\lambda $.

4. Although the paper makes a significant contribution to improving the generalization ability of CLIP and related VLMs, all experiments are conducted solely on the CLIP model (ViT-L/14 and ViT-B/32). The absence of results on other CLIP-like models (e.g., OpenCLIP, EVA-CLIP) may limit the contribution of the proposed methods.

**Questions:**

Please refer to the questions raised in the Weaknesses section above

---

### Official Review · Reviewer_qgzw · 2025-10-29

**Soundness:** 2
**Presentation:** 3
**Contribution:** 3
**Rating:** 4
**Confidence:** 4

**Summary:**

The paper tests whether recent "test-time defenses" for CLIP are truly robust against adversarial attacks. The authors show that robustness is often overestimated when attacks are oblivious to the defense. Tested 6 defense methods. They then propose a simple template, CLIP‑DDT, to build adaptive attacks by targeting what they call each defense’s “key distributional‑difference” step.

**Strengths:**

. Many test‑time defenses were not evaluated with strong defense‑aware attacks; this paper fills that gap.

. Attack the defense’s key step is easy to understand and implement.

. Timely to the community.

**Weaknesses:**

. The paper claims that all test-time defenses measure a "detectable distributional difference" between adversarial and clean data, and this is exploitable, but it does not define it mathematically or show that it is necessary for test‑time defenses. The six “key steps” look quite different (gradient stability, likelihood, Gaussian‑noise consistency, augmentation stability, batch entropy). Can you formalize what constitutes a "distributional difference measurement"? Is CLIP-DDT actually one unified method, or is it 6 different attacks?

. In CLIPure Attack,  why does static attack work better than full gradient in Tables 2–3? This must be investigated (gradient norms, variance, step size/momentum sensitivity, bug checks). In Table 3, improvements over static are tiny. This suggests static attack already breaks the defense.

. In AOM Attack, CLIP‑DDT appears to reduce to a basic full‑gradient attack. Why present this as CLIP-DDT when it's just a full gradient? This weakens their overall claim that CLIP-DDT provides something beyond basic adaptive attacks.


. In R-TPT Attack, the paper uses differentiable augmentations instead of AugMix. This changes the defense as R-TPT specifically uses AugMix, so it's attacking a modified version. But there are well-known gradient estimators for non-differentiable operations. Have you verified that your simplified R-TPT retains the same robustness as the original?

. TTC derivation. Eq. (7) (first‑order drift) and Eq. (8) (indicator) need validation: approximation error vs. numerical gradient; μ sensitivity; how gradients behave when the indicator is 0.

. AOM/TAPT are re‑implemented; only generally similar average robustness is stated. Additionally, CLIPure utilises ViT-L/14, whereas others employ ViT-B/32, and ε varies across tables. Please add per‑dataset deltas vs. original papers, and unify (or justify) backbones/ε. Add seeds and variability for the 1,000‑image sampling.

. Only robust accuracy is reported (no attack‑success rate, convergence, or feature‑space plots). These would make the claims stronger.

**Questions:**

Please check the Weaknesses.

---

### Official Review · Reviewer_suuH · 2025-11-02

**Soundness:** 2
**Presentation:** 3
**Contribution:** 2
**Rating:** 4
**Confidence:** 3

**Summary:**

In this paper, the authors discuss weaknesses of modern test-time dense methods and propose a novel approach, dubbed CLIP-DDT, that offers a principled way to exploit these weaknesses and thereby design a successful attack. In a nutshell, the authors claim that most existing efficient test-time dense methods rely on the distributional discrepancy between clean and adversarial examples, which makes them logical targets for more tailored adversarial attacks. Namely, for each of the considered test-time defense methods they propose a variant of the attack that also tries to minimize such a discrepancy for the attacked image. Finally, the authors demonstrate that for most methods—even with adaptive attacks—one can significantly reduce the robustness of these methods, while integrating CLIP-DDT further improves attack success rates. The authors validate the effectiveness of their approach on a set of popular datasets and against many well-known test-time defense methods, showing that their method boosts attack performance.

**Strengths:**

* The paper initiates the study of effective adversarial attacks on test-time dense approaches, addressing a timely and safety-critical issue in modern VLMs.

* The paper is clearly written and easy to follow; the main idea of the proposed method is well explained and intuitive.

* The paper provides a comprehensive experimental evaluation, covering multiple test-time defenses and popular benchmarks.

**Weaknesses:**

* The authors describe their major contribution as an observation of a common weakness across many popular test-time dense VLM approaches, namely the fact that they rely significantly on the distributional difference between adversarial and clean data. However, in the experimental section, they have to derive specifically designed losses for each of the considered approaches, which feels a bit ad hoc and makes the approach look less general.

* To some extent, the proposed CLIP-DDT method might be viewed as a form of adaptive attack. Therefore, the comparison with “Static” and “Full” is not entirely clear, and the conclusion that using CLIP-DDT for the attacks improves the performance even further over the "Full" attack is not very convincing. One can argue that with better choice of adaptive attack the performance would be different.

* Though acceptable, the paper focuses on a particular type of defenses and their analysis (for example, training-time approaches are not considered in the evaluation), which makes the evaluation to some extent incomplete, as one cannot adequately assess the limits of existing VLM defenses in general and therefore estimate how efficient the proposed attack might actually be against them. As the authors claim in the related work, train-time defenses might reduce the model’s generalization and therefore its performance, which would be good to observe in the evaluation to better understand the trade-off.

**Questions:**

* Since each test-time defense in your experiments requires a specifically designed loss, how general is the proposed CLIP-DDT approach? Could it be applied without such per-method customization / formalized in more generic terms or shared loss function?

* The evaluation focuses on test-time defenses only, and the comparison with the adaptive (“Full”) attack seems limited. Could the authors clarify whether the observed improvement of CLIP-DDT might stem from the suboptimal baseline adaptive attack, rather than from a fundamentally stronger attack strategy?

* The paper does not evaluate training-time defenses, even though they are mentioned in the related work as potentially reducing model generalization. Could the authors comment on how CLIP-DDT would perform against such defenses and whether similar vulnerabilities would still appear in that setting?

---

### Meta-Review · Area_Chair_JmKH · 2026-01-01

**Summary:**

The paper studies the robustness of test-time adaptive adversarial defense methods for CLIP-based VLMs, highlighting potentially overlooked weaknesses that can be exploited by a range of adaptive attacks.  This is an important topic for AI safety, with the increasing reliance on AI systems across a range of critical and emerging use cases.

Reviewers challenged the main premise of a distribution shift underlying existing methods, which is the basis of the proposed zero-shot measurement - since no theoretical justification was presented, and the analysis was focused on a particular type of defense.

In addition, a number of concerns over various technical details and other aspects of the experiment design point to potential issues in the execution and conclusions drawn.

While I acknowledge the significant effort by the authors to prepare such an extensive and important piece of work, the reviews suggest holding off publication till those concerns are addressed.

**Reviewer Concerns:**

Reviewers shared a few key concerns:
- Lack of theoretical justification of the distribution shift underlying existing test-time defenses, which also shows in desperate losses derived for each approach.
- Incomprehensive analysis, either due to focusing on the CLIP family of VLMs or not considering training-time approaches.
- A number of technical details and other aspects of the experiment design (esp by reviewers qgzw and K95W)

**Reviewer Scores:**

Given a considerable overlap in the issues highlighted by the reviewers, and their convergence on a score of 4, this is likely what the final score would be.  While a discussion may help clarify some confusions, in this case it would more likely highlight specific concerns not emphasized in all reviews.

Unfortunately, no rebuttal was submitted for any of the reviews, leaving those concerns unresolved.

---

### Decision · Program_Chairs · 2026-01-26

Reject